# Proper Selection of In Vitro Cell Model Affects the Characterization of the Neutralizing Antibody Response against SARS-CoV-2

**DOI:** 10.3390/v14061232

**Published:** 2022-06-07

**Authors:** Elena Criscuolo, Benedetta Giuliani, Davide Ferrari, Roberto Ferrarese, Roberta A. Diotti, Massimo Clementi, Nicasio Mancini, Nicola Clementi

**Affiliations:** 1Laboratory of Microbiology and Virology, Vita-Salute San Raffaele University, 20158 Milan, Italy; criscuolo.elena@hsr.it (E.C.); giuliani.benedetta@hsr.it (B.G.); ferrarese.roberto@hsr.it (R.F.); diotti.robertaantonia@hsr.it (R.A.D.); clementi.massimo@hsr.it (M.C.); mancini.nicasio@hsr.it (N.M.); 2SCVSA Department, University of Parma, 43121 Parma, Italy; davide.ferrari@unipr.it; 3IRCCS Ospedale San Raffaele, 20158 Milan, Italy

**Keywords:** COVID-19 vaccine, BNT162b2 mRNA vaccine, SARS-CoV-2, neutralizing activity, antibody response, VOCs

## Abstract

(1) Background: Our aim is the evaluation of the neutralizing activity of BNT162b2 mRNA vaccine-induced antibodies in different in vitro cellular models, as this still represents one of the surrogates of protection against SARS-CoV-2 viral variants. (2) Methods: The entry mechanisms of SARS-CoV-2 in three cell lines (Vero E6, Vero E6/TMPRSS2 and Calu-3) were evaluated with both pseudoviruses and whole virus particles. The neutralizing capability of sera collected from vaccinated subjects was characterized through cytopathic effects and Real-Time RT PCR. (3) Results: In contrast to Vero E6 and Vero E6/TMPRSS2, Calu-3 allowed the evaluation of both viral entry mechanisms, resembling what occurs during natural infection. The choice of an appropriate cellular model can decisively influence the determination of the neutralizing activity of antibodies against SARS-CoV-2 variants. Indeed, the lack of correlation between neutralizing data in Calu-3 and Vero E6 demonstrated that testing the antibody inhibitory activity by using a single cell model possibly results in an inaccurate characterization. (4) Conclusions: Cellular systems allowing only one of the two viral entry pathways may not fully reflect the neutralizing activity of vaccine-induced antibodies moving increasingly further away from possible correlates of protection from SARS-CoV-2 infection.

## 1. Introduction

From the very beginning of the SARS-CoV-2 pandemic, great efforts have been made to elucidate the viral lifecycle and virus–host interactions. In this regard, the study of the two entry pathways exploited by the virus to infect susceptible cells can provide insights into the tropism of SARS-CoV-2 variants as well as possible novel diagnostic and antiviral approaches. Upon the interaction between the Receptor Binding Site (RBD) of the Spike (S) protein and the Angiotensin-Converting Enzyme 2 (ACE2), different host-cell proteases can cleave the S protein at the surface of the infected cells or into the endosomal compartments, distinguishing two viral entry mechanisms [1,2,3]. On the cell surface, transmembrane serine protease 2 (TMPRSS2) proteolytic cleavage exposes the fusion peptide domain of the S protein, allowing a direct fusion between viral and host cell membranes [3,4,5,6]. Alternatively, cathepsin L inside the phagolysosome activates the S protein, inducing the fusion process [3,7].

Due to its central role in both viral entry and its immunogenicity [8], the S protein has been carefully monitored over time from the very beginning of the COVID-19 pandemic. Through genome sequencing analysis, different variants of concern (VOCs) and variants of interest (VOI) have been identified, and different capabilities to exploit the two entry processes have been observed [9,10,11]. This could impact both the cell tropism and the transmissibility, explaining the differences in the spread of the viral variants all over the world [10].

The key role of the S glycoprotein in virus entry and its capability to elicit antibodies hampering cell infection enlightens why its sequence was used to develop COVID-19 vaccines. As the serum-neutralizing capability represents, thus far, one of the best surrogates of protection from COVID-19, the possible escape of novel VOCs from antibodies elicited by vaccines, has raised concerns [12]. To date, the gold standard model to evaluate the neutralizing activity of antibodies are Vero E6 cells, even if these cells do not express TMPRSS2 on their surface. This aspect should be considered to correctly recapitulate the complexity of the viral entry mechanisms exerted by SARS-CoV-2 in vivo. Thus, the differences between the in vitro model and in vivo infection could also negatively affect the correct evaluation of one of the few correlates of protection readily quantifiable for both diagnostic and prognostic purposes.

In this regard, this project aims to demonstrate possible discrepancies in the neutralization assay using different cells against VOCs of epidemiological interest.

## 2. Materials and Methods

### 2.1. The Clinical Samples

The Covidiagnostix is a multicenter study, approved by the San Raffaele Hospital, Milan, Italy, Institutional Ethical Review Boards (CE:199/INT/2020), which aims to monitor the antibody response of a population of healthcare professionals (HCPs) who were offered the BNT162b2 mRNA COVID-19 (Comirnaty) vaccine [13]. This study included 1052 HCPs from the San Raffaele Hospital, Milan, Italy. All HCPs received two doses of the BNT162b2 vaccine (21 ± 1 day interval between the two doses) during January and February 2021; no exclusion criteria were applied. Blood samples were withdrawn for serological evaluation, as previously described [14].

Eight serum samples were selected from BNT162b2 COVID-19 (Comirnaty^®^) eight vaccinated subjects, 21 days after receiving the second dose (Appendix A). Subjects #2 and #5 had a clinical history of SARS-CoV-2 infection before being vaccinated, while the others were never infected by SARS-CoV-2. Sera were selected based on their previous characterization using the electrochemiluminescence immunoassay (ECLIA) Roche Anti-SARS-CoV-2-S test (Roche, Basel, Switzerland), which detects pan-immunoglobulins (IgA, IgG and IgM) against the S receptor binding domain (RBD): four samples had a high anti-RBD titer (#2, #5, #17 and #26), two intermediate titers (#32 and #25) and two with a low titer (#29 and #37) [13].

### 2.2. Cell Lines and Viruses

Vero E6 (Vero C1008, clone E6; ATCC CRL-1586) cells were cultured in Dulbecco’s modified Eagle’s medium (DMEM) supplemented with non-essential amino acids (NEAA), penicillin/streptomycin (P/S, 100 U/mL), HEPES buffer (10 mM) and 10% (*v*/*v*) heat-inactivated fetal bovine serum (FBS). We added 1 mg/mL of Geneticin (G418) to medium of Vero E6 that stably expressed TMPRSS2 (Vero E6/TMPRSS2, NIBSC 100978). Calu-3 (Human lung cancer cell line, ATCC HTB-55) cells were cultivated in Minimum Essential Medium (MEM) supplemented with NEAA (1×), P/S (100 U/mL), 1 mM sodium pyruvate and 10% (*v*/*v*) heat-inactivated FBS. All cell lines were incubated at 37 °C and 5% CO_2_ in a humidified atmosphere. All cell lines were routinely tested for mycoplasma (Lonza, LT07-218).

Six clinical isolates of SARS-CoV-2 were obtained and propagated in Vero E6 and in Vero E6/TMPRSS2 cells: D614G (GSAID accession ID: EPI_ISL_413489), Alpha (GSAID accession ID: EPI_ISL_1924880), Beta (GSAID accession ID: EPI_ISL_1599180), Gamma (GSAID accession ID: EPI_ISL_1925323), Delta (GSAID accession ID: EPI_ISL_4198505) and Omicron BA1 (GSAID accession ID: EPI_ISL_12188061).

In detail, 0.8 mL of the transport medium of the nasopharyngeal swab (COPAN’s kit UTM^®^ universal viral transport medium—COPAN) was mixed 1:1 with DMEM without FBS and supplemented with P/S and Amphotericin B. The mixture was added to an 80% confluent Vero E6 cell monolayer seeded in a 25 cm^2^ tissue culture flask. After 1 h adsorption at 37 °C, 3 mL of DMEM supplemented with 2% FBS and Amphotericin B was added. One day post-infection (dpi), the monolayer was washed in PBS, and 4 mL of DMEM supplemented with 2% FBS and Amphotericin B was added. The cytopathic effect was monitored using inverted phase-contrast microscopy (Olympus CKX41), and the supernatant was collected at monolayer complete disruption (3 dpi). 

The sample was heat-inactivated at 56 °C for 30 min, and the viral genome was extracted using a QIAamp Viral RNA Mini Kit following the manufacturers’ instructions. h eextracted RNA was processed with the CleanPlex^®^ SARS-CoV-2 Panel (Paragon Genomics, Hayward, CA, USA) and sequenced with the MiSeq Reagent Kit v2 (300-cycles) (Illumina, San Diego, CA, USA) on the MiSeq platform. Genomic reconstruction was performed using the SOPHiA DDM™ platform (SoPHiA Genetics, Boston, MA, USA).

### 2.3. Virus Titration

Virus stocks were titrated using an Endpoint Dilutions Assay (EDA, TCID_50/mL_). Vero E6 cells were seeded into 96-well plates and infected at 95% of confluency with base 10 dilutions of virus stock. After 1 h of adsorption at 37 °C, the cell-free virus was removed, cells were washed with PBS 1×, and a complete medium was added to cells. After 72 h, cells were observed to evaluate the presence of a cytopathic effect (CPE). TCID_50/mL_ of viral stocks were then determined with the Reed–Muench formula.

### 2.4. Pseudovirus Generation

We generated lentiviral pseudoviruses following the protocol already described [15]. Briefly, 10^6^ HEK-293T cells were seeded in a 6-well plate, and 24 h later, were co-transfected using Lipofectamine 2000 (Invitrogen, Waltham, MA, USA) with five plasmids (BEI Resources Repository): pHAGE with CMV-driven Luciferase-IRES-ZsGreen (NR-52516), pHDM with HIV Gag-Pol (NR-52517), pHDM with HIV Tat (NR-52518), pRC with CMV-driven HIV Rev (NR-52519), pHDM-Spike D614G C-term 21 bp deletion (NR-53765). At 24 h post-transfection, the medium was changed with pre-warmed DMEM supplemented with 2% of FBS and incubated at 37 °C and 5% CO_2_. At 60 h post-transfection, cell supernatant was harvested and filtered through 0.45 μm filter (Millipore, Burlington, MA, USA) to eliminate cell debris. Pseudoparticles in the media were subsequently pelleted by ultracentrifugation through a 20% sucrose cushion at 26,000 rpm for 3 h by using Beckman 328 SW28 rotor. Pseudoviruses were aliquoted and stored at −80 °C.

### 2.5. Pseudovirus Titration

To determine pseudovirus titers, we used a luciferase assay. In detail, 4 × 10^5^ cells/mL were seeded in 96-well plates and cultured at 37 °C and 5% CO_2_ to be confluent after 24 h. The pseudovirus dilutions (toto and 1:2 serial dilutions) were added to the target cells, and spinoculation (1 h at 800 g) followed to allow pseudovirus adsorption. Subsequently, media was removed, and fresh medium was added to cells. The cell supernatant was removed 72 h post-infection, and the cells were lysed with 100 μL of Glo Lysis Buffer (Promega, Madison, WI, USA) for 15 min at room temperature. Cell lysates were transferred to a luminometer plate and 100 μL of Bright-Glo Assay Reagent (Promega) was added immediately before detection (Victor3, Perkin Elmer, Waltham, MA, USA). For further assays, we selected the dilution in which the Relative Luminescence Units (RLUs) were sufficiently (>1000-fold) above the background.

### 2.6. XTT Assay for Determination of Cell Viability

Cell viability assays were performed using the Cell Proliferation kit II (XTT) (Roche Diagnostics, Merck, Darmstadt, Germany). Briefly, the tetrazolium salt 2,3-bis-(2-methoxy-4-nitro-5-sulfophenyl)-2H-tetrazolium- 5-carboxanilide (XTT) was cleaved by viable cells to form an orange formazan dye that can be quantified photometrically at 450 nm. Before the assay, Vero E6/TMPRSS2 cells (4 × 10^5^ cells/mL) were cultured in 96-well plates for 24 h. The culture medium was replaced by medium containing the inhibitors, and cells were incubated for 72 h. XTT was added to each well, and the plates were incubated for an additional 2 h. The optical density was measured at 450 nm (reference wavelength—650 nm) using a Multiskan GO plate reader (Thermo Scientific Instruments, Waltham, MA, USA). For quantifications, the background levels of media without cultured cells were subtracted.

### 2.7. Inhibition of Pseudovirus Entry

Target cells (4 × 10^5^ cells/mL) were seeded in 96-well plates and cultured for one day at 37 °C and 5% CO_2_. Cells were pre-treated with Bafilomycin A1 (100 nM, BFLA-1, Merck), camostat mesylate (100 μM, Merck), alone or in combination, 24 h before the transduction with pseudovirus. Cells were centrifuged at 800 g for 1 h at 37 °C and subsequently incubated for 3 days at 37 °C and 5% CO_2_. Cells were then lysed with 100 μL of Glo Lysis Buffer (Promega) for 15 min at room temperature. Cell lysates were transferred to a luminometer plate, and 100 μL of Bright-Glo Assay Reagent (Promega) was added immediately before detection (Victor3, Perkin Elmer). We performed six biological replicates for each condition.

### 2.8. Inhibition of SARS-CoV-2 Entry

Target cells (4 × 10^5^ cells/mL) were seeded in 96-well plates and cultured for one day at 37 °C and 5% CO_2_. Cells were pre-treated with either one or the combination of BFLA-1 (100 nM) and camostat mesylase (100 μM) 1 h before the infection. Cells were then infected for 1 h with D614G virus variant (0.001 MOI). After virus adsorption, cells were washed three times with PBS and incubated with medium containing the inhibitors and supplemented with 2% FBS. To evaluate the infection inhibition capability of the chemical compounds in Vero and Vero E6/TMPRSS2 cells, we estimated the presence of CPE 48 h post infection (hpi). 

The following scoring system was used: 0 = uninfected; 0.5 to 2.5 = increasing number/area of plaques; 3 = all cells infected. Infection control (score 3) was set as 0% infection inhibition, uninfected cells (score 0) as 100% infection inhibition. The whole surface of the wells was considered for the analysis (5× magnification) in inverted phase-contrast microscopy (Olympus CKX41). 

In Calu-3 experiments, the supernatants were collected at 48 hpi, and the relative copy number of viral genomes were evaluated using Real Time RT-PCR. For the analysis of the viral entry mechanisms of the selected VOCs, Calu-3 cells were pre-treated with BFLA-1 and camostat mesylate 1 h before the infection (0.1 MOI). Medium with the same concentration of chemical compounds was replaced 24 hpi. The cell supernatants were collected at 72 hpi and analyzed as described above. All the experimental conditions were performed in triplicate.

### 2.9. Viral RNA Extraction and Real-Time RT-PCR

Viral RNA was purified from cell culture supernatant using the QIAamp Viral RNA Mini Kit (QIAGEN, Hilden, Germany), following the manufacturer’s instructions. Subsequently, the purified RNA was used as template to synthesize the first-strand cDNA, using the SuperScript™ First-Strand Synthesis System for RT-PCR (Thermo Fisher Scientific, Waltham, MA, USA), following the manufacturer’s instruction. Real-time PCR, using SYBR^®^ Green dye-based PCR amplification and detection method, was performed to detect the cDNA. 

We used the SYBR™ Green PCR Master Mix (Thermo Fisher Scientific, Waltham, MA, USA), the forward primer N2F: TTA CAA ACA TTG GCC GCA AA, the reverse primer N2R: GCG CGA CAT TCC GAA and the following PCR conditions: 95 °C for 2 min, 45 cycles of 95 °C for 20 s, annealing at 55 °C for 20 s and elongation at 72 °C for 30 s, followed by a final elongation step at 72 °C for 10 min [16]. RT-PCR was performed using the ABI-PRISM 7900HT Fast Real-Time instruments (Applied Biosystems, Waltham, MA, USA) by using optical-grade 96-well plates. Samples were run in duplicate in a total volume of 20 μL.

### 2.10. Real-Time qPCR Analysis of ACE2 and TMPRSS2 Expression Levels

Cellular RNA from Vero E6, Vero E6/TMPRSS2 and Calu-3 (2 × 10^6^ cells) were extracted using the RNeasy Mini Kit (QIAGEN) according to the manufacturer’s protocol. Then, the mRNA from each sample was reverse transcribed using the SuperScript™ III First-Strand Synthesis System for RT-PCR (Thermo Fisher Scientific, Waltham, MA, USA), following the manufacture’s instruction. We analyzed 10 ng of cDNA to evaluate the expression levels of ACE2 and TMPRSS2 with Real Time RT PCR using the SYBR^®^ Green dye-based PCR amplification and detection method. 

Gene-specific primers for human ACE2 (FW: AAA CAT ACT GTG ACC CCG CAT; RE: CCA AGC CTC AGC ATA TTG AAC A), monkey ACE2 (FW: AAA CAT ACT GTG ACC CCG CAT; RE: GCT TCA GCA TAT TGA GCA ATT TCT G) and human TMPRSS2 (FW: AAT CGG TGT GTT CGC CTC TAC; RE: CGT AGT TCT CGT TCC AGT CGT). As endogenous control, we used β-actin (FW: CCC TGG ACT TCG AGC AAG AG; RE: ACT CCA TGC CCA GGA AGG AA). Amplification was performed under the following conditions: 94 °C for 5 min, 45 cycles of 94 °C for 30 s, annealing at 60 °C for 30 s and elongation at 68 °C for 30 s, followed by a final elongation step at 72 °C for 10 min. Samples were run in triplicate in a total volume of 20 μL.

### 2.11. Western Blot Assay

Vero E6, Vero E6/TMPRSS2 and Calu-3 cells (10^6^ cells) were detached with TrypLE (Thermo Fisher, Waltham, MA, USA), pelleted at 700g for 5 min and lysed in RIPA buffer (Thermo Scientific, Waltham, MA, USA) supplemented with an EDTA-free protease inhibitor cocktail tablet (Thermo Scientific, Waltham, MA, USA) for 30 min at 4 °C. Then, lysed cells were clarified at 1000 rpm for 5 min at 4 °C. 10 μL of protein samples were added to 4× LDS sample buffer (Thermo Fisher Scientific, Waltham, MA, USA), loaded into a Bolt 4–12% Bis-Tris Gel (Thermo Fisher Scientific, Waltham, MA, USA) and electrophoresed by SDS-Page at 200 V for 40 min in MES 1× buffer (Thermo Fisher Scientific, Waltham, MA, USA). Proteins were transferred to a polyvinylidene difluoride (PVDF) membrane at 4 °C for overnight in ice-cold Western Transfer Buffer (25 mM Tris, 192 mM Glycine, MeOH 20% (*v*/*v*)). The membrane was blocked with 5% BSA in TBS containing 0.1% Tween-20 (PBS-T) for 1 h. 

Next, the membrane was incubated for at least 1 h with primary antibodies specific for TMPRSS2 (Abcam, ab109131; diluted 1:1000) and β-actin (Abcam, ab8227; diluted 1:1000). The membranes were washed 3 times in PBS-T, followed by probing with horseradish peroxidase (HRP)-conjugated anti-rabbit antibody (A0545, Merck; diluted 1:4.000) and anti-mouse (A4416, Merck; diluted 1:8.000) as secondary. Signal was developed by treating membranes with SuperSignal West Pico Chemiluminescent Substrate (Thermo Fisher Scientific, Waltham, MA, USA) imaging on a ChemiDoc MP System (Bio-Rad #12003154, Hercules, CA, USA).

### 2.12. Immunofluorescence Assays

Vero E6, Vero E6/TMPRSS2 or Calu-3 were seeded on Matrigel^®^ coated slides with a removable 12-well silicone Chamber (Ibidi). After 24 h, the cells were fixed and permeabilized with ice-cold methanol-acetone (1:1) for 15 min at room temperature. Cells were stained with the primary antibody for 1 h at 37 °C: rabbit pAb anti-ACE2 (Sino Biologicals, Beijng, China, 10108-T24) or rabbit pAb anti-TMPRSS2 (Sino Biologicals, 204314-T08). Then, the secondary antibody was added: goat anti-rabbit IgG Alexa Fluor 488 (Invitrogen, A-11008). Cell nuclei were stained with Hoechst 33342 (Thermo Fisher Scientific, Waltham, MA, USA). The images were acquired with Zeiss Axio Observer.Z1 microscope with QImaging Exi-Blue (Carl Zeiss, Oberkochen, Germany) at 20-fold magnification.

The evaluation of the receptor expression was performed with ImageJ software. We calculated the ratio between the total green fluorescent signal and the number of nuclei for both receptors. Then, we normalized these values on reference cell line (Calu-3 cells).

### 2.13. Kinetic Profiles

Vero E6, Vero E6/TMPRSS2 and Calu-3 (3 × 10^5^ cells/mL) cells were seeded in 96-well plates and cultured for 1 day at 37 °C and 5% CO_2_ in a humidified atmosphere. Then, the cells were infected with the different SARS-CoV-2 variants (0.001 multiplicity of infection, MOI) in triplicate. After 1 h of adsorption, cells were washed three times with PBS to remove cell-free virus, and fresh medium was added. Cell supernatants were collected at 6 time points: 1, 3, 6, 24, 48 and 72 hpi, and viral genomes were extracted and analyzed by Real-Time RT-PCR.

### 2.14. Microneutralization Experiments

Cells were seeded into 96-well plates 24 h before the experiment performed at 95% cell confluency for each well. Serum samples were decomplemented by incubation at 56 °C for 30 min, and two dilutions (1:80 and 1:160) were incubated with SARS-CoV-2 at 0.01 MOI for 1 h at 37 °C. Virus–serum mixtures and positive infection control were applied to cells monolayers after a PBS 1× wash, and virus adsorption was performed at 37 °C for 1 h. 

Then, the cells were washed with PBS 1× to remove cell-free virus particles and virus-containing mixtures and controls were replaced with complete medium supplemented with 2% FBS. The plates were incubated at 37 °C in the presence of CO_2_ for 72 h. The experiments were performed in triplicate. Neutralization activity was evaluated by comparing CPE presence detected in the presence of virus–serum mixtures to positive infection control (Vero and Vero E6/TMPRSS2 cells), or by Real-Time RT PCR (Calu-3 cells).

### 2.15. Statistical Analysis

Two-way ANOVA and Sidak’s multiple comparisons were performed to analyze the RLUs obtained from pseudovirus particles experiments with BFLA1 and camostat. When using BFLA1 and camostat against the whole virus, CPE observed for different experimental settings in Vero E6 cells using, alone or in combination, were normalized to corresponding virus infection control. 

For Calu-3 experiments, Real Time-PCR results were analyzed calculating Delta (Δ) Ct as the difference between Ct values obtained for experimental settings and infection control. In both cases, two-way ANOVA and Tukey’s multiple comparisons were performed to analyze the results. ΔCt was calculated also in kinetics profiles experiments with Calu-3 cells as the difference between Ct values obtained for the different time points and Ct(1 h).

Two-way ANOVA and Tukey’s multiple comparisons were performed for both the comparison of the different time points and the general trends in the different cell lines. Sidak’s multiple comparisons was performed to analyze the BFLA-1 and camostat mesylate inhibitory activity against the VOCs. Two-way ANOVA and Dunnett’s multiple comparisons were performed for XTT cell viability evaluation. All the analyses were performed on GraphPad Prism 8.

## 3. Results

### 3.1. Evaluation of Expression Levels of Host Molecules Involved in Viral Entry Process

We focused our study on three cell lines: Vero E6, Vero E6/TMPRSS2 and Calu-3. First, we assessed the gene expression of ACE2 and TMPRSS2 (Figure 1A). The quantification of mRNA levels showed that Vero E6 and Vero E6/TMPRSS2 presented 34.2 (±4.4) and 333.4 (±179.8)-fold higher levels of ACE2 than Calu-3. Vero E6 did not express TMPRSS2, as expected, while transfected Vero E6 showed 481.1 (±72.1)-fold higher level of the TMPRSS2 than Calu-3. To analyze the TMPRSS2 protein expression, we performed a Western Blot analysis (Appendix A), which had extremely variable and unprecise quantification due to the low sensitivity of the monoclonal antibodies in the experimental assay.

The detected discrepancies in the mRNA expression levels of receptors between the cell lines is in accordance with the IF analysis. Indeed, the investigation of the protein levels showed that Vero E6 and Vero E6/TMPRSS2 cells presented 2.1-fold higher expression of ACE2 than the reference cells (Figure 1B,C). Last, Vero E6 cells did not express TMPRSS2, whereas Vero E6/TMPRSS2 displayed 4.9-fold higher levels of the protein compared to Calu-3.

### 3.2. Pseudovirus Entry in the Different Cell Lines

To assess how SARS-CoV-2 could enter different cell lines, we generated pseudotyped viral particles, as they represent a safe and convenient assay system for studying the entry processes. Pseudovirus-based assays have been widely used for the study of cellular tropism, receptor recognition and viral inhibitors as well as the evaluation of neutralizing antibodies [17].

To dissect the two viral entry mechanisms, we used BFLA-1 and camostat mesylate, which selectively hamper endocytosis and the direct fusion with the plasma membrane, respectively. In our system, we observed that pseudoparticles entered the target cells with very different efficiency between Vero E6 and Vero E6/TMPRSS2 cells (Appendix A). Consequently, to obtain consistent results using the entry inhibitors in Vero E6 cells, we used 10 times the concentration of pseudoviruses that in engineered cells (Appendix A). 

The analysis showed that BFLA-1 similarly inhibited the two cell lines (60.4 ± 10.1% in Vero E6, 51.5 ± 18.4% in Vero E6/TMPRSS2), while the camostat mesylate blocked the entry more efficiently in the engineered cells, both when used alone (Vero E6/TMPRSS2: 78.2 ± 11.8%, Vero E6: 47.3 ± 35.5%, *p* < 0.05) or when combined with BFLA-1 (Vero E6/TMPRSS2: 99.9 ± 0.04%, Vero E6: 73.9 ± 8.4%, *p* < 0.05) (Figure 1D). These results were consistent with the characteristics of the two cell lines observed during the ACE2 and TMPRSS2 expression analysis, yet they showed that comparison of the three selected cell lines using the same amount of virus was not feasible. We then decided to discontinue the pseudovirus-based assay and use the clinical isolates for the subsequent experiments.

### 3.3. Virus Entry in the Different Cell Lines

The same evaluation previously described using pseudoviral particles was performed with a clinical isolate to characterize the entry mechanisms in the three selected cell lines. We observed that BFLA-1 treatment fully protects Vero E6 from virus infection (100%), confirming that SARS-CoV-2 could enter this cellular model by exploiting only the endocytic pathway (Figure 1E). In contrast, BFLA-1 partially reduced the infection in Calu-3 (45.6 ± 1.8%, *p* < 0.0001) and showed no protective effect on Vero E6/TMPRSS2 cells (0%, *p* < 0.0001). 

To inquire into the role of S protein priming by TMPRSS2 activity in the different in vitro models of infection, cells were treated with camostat mesylate. As expected, the serine protease inhibitor did not hamper the viral entry process in Vero E6 cells since the virus could only exploit the endocytic process. The treatment remarkably inhibited the entry of SARS-CoV-2 in Calu-3 cells (89.1 ± 2%, *p* < 0.0001), whereas no inhibition (0%) was observed in Vero E6/TMPRSS2 cells. It must be noted that, in engineered Vero E6, the highest nontoxic concentration of this compound did not interfere with the virus entry (Appendix A).

Finally, the combination of the two inhibitors led to complete protection from viral infection only in Calu-3 cells (99.7 ± 0.6%, *p* < 0.0001) whereas the combination of the two inhibitors resulted in an intermediate protective effect (33.3%) on Vero E6 cells.

### 3.4. Replication Kinetics of SARS-CoV-2 Variants in the Different Cell Lines

We investigated if the differences in the entry pathways observed in the three cell lines could affect the replication kinetics of SARS-CoV-2 viral variants. After testing D614G as well as five VOCs, the results showed that only the growth curves of the Alpha variant did not differ between the cell lines at the tested time points (Figure 2). In detail, the D614G variant curve in Vero E6/TMPRSS2 cells started to differ from those obtained on the other cultures at 24 hpi (*p* < 0.0001), while Calu-3 and Vero E6 diverge only at 72 hpi (*p* < 0.0001). 

The replication of Beta, Delta and Omicron variants in Calu-3 was different from the other two cell lines starting from 3 hpi (Beta: *p* < 0.0001 vs. Vero E6 and *p* < 0.001 vs. Vero E6/TMPRSS2; Delta: *p* < 0.01 vs. Vero E6, *p* < 0.05 vs. Vero E6/TMPRSS2; Omicron: *p* < 0.0001 vs. Vero E6 and Vero E6/TMPRSS2), while results obtained from Vero E6 and Vero E6/TMPRSS2 cultures were different 24 hpi for all the three virus variants infections (*p* < 0.0001). 

However, we observed that results from Calu-3 and Vero E6 experiments with Beta and Omicron infections came closer at the last time point (72 hpi, *p* < 0.05). The data obtained from the analysis of the Gamma infection of Calu-3 cells were the only data that diverged from the other two cells starting from 6 hpi (*p* < 0.0001), and the differences between wild type and engineered Vero E6 cells were detected already at 3 hpi (*p* < 0.001) and became more statistically significant from 24 hpi (*p* < 0.0001).

Considering the replication trends of the different VOCs in the selected cell lines, the heatmaps highlighted how the presence of the viral genome in the culture media strongly increased starting from 24 hpi for all viral variants, except Omicron, which was already detectable a few hours post-infection (Figure 3). The heatmaps also confirmed how Alpha variant replication was comparable in the three cellular models, while the overall trend of the other variants in the three cell lines was always statistically different (Figure 3). 

Interestingly, D614G kinetics showed a minor difference between engineered Vero E6 and Calu-3 (*p* < 0.01) than between Vero E6 and the other two cells (*p* < 0.0001), while the slightest difference was observed in Beta, Gamma and Omicron replications comparing Vero E6 to Calu-3 cells (*p* < 0.05), suggesting that these viral variants may exploit the two entry pathways one the opposite of the other. Delta variant replication in Vero E6 cells was the slowest observed.

### 3.5. Evaluation of Viral Variants Entry Mechanisms in Calu-3 Cells

To assess the possible implications of the S protein mutations in the viral entry process, we analyzed the capability of either BFLA-1 or camostat mesylate to inhibit viral infection in Calu-3 cells, as the virus infects these cells using both entry mechanisms (Figure 4).

In detail, Gamma and Alpha variants showing infection inhibition equal to 86.12 ± 5.24 and 80.73 ± 4.28%, respectively, were the most affected by the presence of camostat mesylate suggesting the central role of the TMPRSS2 activity in priming the S protein and inducing the fusion between the viral envelope and cellular plasma membrane. This compound also hampered the infection from Beta (44.09 ± 4.90%) and Delta (33.10 ± 4.16%) variants yet showed little inhibitory effect on the entry process of the Omicron variant (18.01 ± 2.68%).

All the analyzed clinical isolates also exploited the endocytic pathway to infect Calu-3 cells as suggested by the BFLA-1 data. The compound affected the entry process of all viral variants (Alpha: 32.36 ± 3.16%, Beta: 27.60 ± 3.30%, Gamma: 68.61 ± 4.26%, Delta: 18.02 ± 5.71%, Omicron: 22.64 ± 1.13%). Interestingly, Omicron infection was equally hampered by both chemical compounds. Except for Omicron, the other clinical isolates showed that interference with the TMPRSS2 activity led to a significantly higher protective effect than what observed with BFLA-1 (*p* < 0.0001).

### 3.6. Neutralizing Activity of Vaccinated Subjects’ Sera

Primary sequence modification in S protein may be implicated in the reduced sensitivity of SARS-CoV-2 variants to the humoral immune response elicited by vaccination. To evaluate this hypothesis, we tested eight sera from vaccinated subjects three weeks after receiving the second dose of Pfizer-BioNTech COVID-19 Vaccine BNT162b2 (Comirnaty^®^) against four selected variants. Delta variant was excluded from this study because of its difficulty to infect efficiently Vero E6 cells generate viral stocks with high titer.

The results obtained on Vero E6 cells showed that almost all the selected sera efficiently neutralized the D614G variant (1:80 sera dilution: 88.89 ± 18.59%, 1:160 sera dilution: 81.48 ± 15.18%), and the neutralizing activity gradually decrease when the Alpha, Beta and Omicron variants were tested, as the more effective sera (#2 and #5) dropped to 50% of infection inhibition (Figure 5). The assays performed using engineered Vero E6 cells showed an even more marked reduction in activity against the different variants, culminating in a zero inhibition of the serums against Omicron. 

However, sera #2 and #5 remained as those with the most marked activity among the tested cohort. Interestingly, with respect to Vero E6 and engineered Vero E6, Calu-3 showed a similar or higher neutralizing activity against all variants except Omicron (Figure 5) for which the neutralizing activity was minimal, with only one serum (#2) able showing activity equal to 50%.

Finally, we examined if the limitations of the Vero E6 gold standard model could impact the characterization of neutralizing antibody response by correlating the infection inhibition results obtained in Vero E6 and Calu-3 (Figure 6). The Spearman’s correlation coefficient indicated an almost perfect correlation between the infection inhibition percentages at 1:80 sera dilution against the Alpha variant in the cellular models (r = 0.95, *p* < 0.001), whereas this value slightly decreased when using a 1:160 sera dilution (r = 0.8, *p* < 0.05). In contrast, no correlation was observed for the other three variants (*p* > 0.05) at any sera dilutions, indicating that neutralizing data obtained in Calu-3 did not significantly correlate with the data obtained in Vero E6.

## 4. Discussion

An in vitro model that resembles in vivo SARS-CoV-2 entry dynamics would be fundamental to dissect both the viral entry process and the neutralizing antibody response elicited by vaccination.

The Vero E6 cell line has been broadly used to study the lifecycle of several viruses, due to its natural dysregulation of the IFN response [17,18]. This represents the most used cell model for SARS-CoV-2 due to the high expression of ACE2 on their surface. However, the absence of TMPRSS2 on the plasma membrane of Vero E6 offers SARS-CoV-2 could only one of the two entry pathways representing a limitation when studying the different entry mechanisms of the SARS-CoV-2 VOCs [3]. 

For this reason, we compared three in vitro models: Vero E6, engineered Vero E6 that stably express TMPRSS2 and Calu-3 cells. Since Calu-3 are human-derived lung adenocarcinoma cells, which is closer to the in vivo target of SARS-CoV-2 infection, they were used as a reference to compare the expression of ACE2 and TMPRSS2 [19].

In accordance with the mRNA level investigation, the IF analysis confirmed the discrepancies between the expression levels of ACE2 and TMPRSS2 between Vero E6, Vero E6/TMPRSS2 and Calu-3. ACE2 levels were remarkably higher in Vero E6 and engineered Vero E6 compared to Calu-3. Our studies confirmed the lack of TMPRSS2 expression in Vero E6 cells, which, in contrast, was overexpressed in engineered Vero E6 (four-fold higher) compared to human-derived cells.

Once assessed the expression of host factors involved in viral entry process, we dissected the entry mechanisms of SARS-CoV-2 in the three selected cell lines. First, we used pseudoviral particles since they only recapitulated the viral entry event [20,21,22,23]. However, we observed important differences between the cell cultures, thus, being an important limit to our aim of comparing the three cell lines. The subsequent use of inhibitors confirmed this imbalance. 

We used inhibitors that selectively interfere with the two viral entry processes, camostat mesylate and BFLA-1. The first targets TMPRSS2, hampering its capability to cleave S protein and preventing the direct fusion between the host plasma membrane and the viral envelope. To interfere with the endocytic pathway, we used BFLA-1, a vacuolar H^+^-ATPase inhibitor. The acid pH environment in the endosomal compartment is fundamental for activating cathepsin L, which is the endosomal protease responsible for the cleavage of S protein and the exposure of the FP domain. Thus, the inhibition of the phagosome-lysosome fusion hampered SARS-CoV-2 entry. 

We observed that, to obtain consistent results, we had to used 10-fold more virus in wild type Vero E6 compared to engineered cells. At the best of our knowledge, no previous studies reported these inconsistences between the capability of pseudoviruses to infect these two cell lines. Due to this significant difference in the pseudoparticles titers, we did not continue the characterization on the Calu-3 and decided to use a clinical isolate for the subsequent experiments. In accordance with previous studies, treating Vero E6 cells with BFLA-1 resulted in complete protection from the infection [3,6,24], suggesting that SARS-CoV-2 could enter this cell line only through the endocytic pathway. In our analysis, the same compound showed no effect on Vero E6/TMPRSS2 cell infection, indicating how the overexpression of the serine protease led to the entry of all viral particles at the plasma membrane level. 

Interestingly, targeting the TMPRSS2 activity in the engineered Vero E6 with camostat mesylate did not interfere with the viral entry in this cellular model. We speculated that the inhibitor concentration used [3,25] was not sufficient to interfere with the activity of the overexpressed TMPRSS2 in the engineered cells. To test our hypothesis, we pre-treated the cells with the highest non-toxic concentration of camostat mesylate; however, it did not block virus entry. 

Therefore, we underlined an unbalance entry dynamic in both cellular models: the endocytic or the direct fusion at the plasma membrane were exploited by SARS-CoV-2 in Vero E6 and Vero E6/TMPRSS2, respectively. The unexpected effect of the combination of the two inhibitors on Vero E6 cells may derive from a non-predicted interaction between the BFLA-1 and camostat mesylate. When we tested the mixture in the pseudovirus assay, which recapitulates only the early steps of viral entry to target cells, we did not observe any decrease in its activity. However, we observed a BFLA-1 power loss when tested in combination with camostat in the more complex environment resulting from the infection with the whole virus. It is possible that, when used in a system that recapitulates all the variables involved in the virus infection, its biological activity result is weaker.

We also treated the Calu-3 cells with the two inhibitors to characterize the entry dynamics. The selective inhibition of the two pathways showed a different degree of protection from the infection, implying that SARS-CoV-2 did not exploit equally the two entry mechanisms. In detail, the interference with TMPRSS2 activity displayed a remarkable inhibitory effect than the one obtained targeting the endosomal acidification. Literature data showed that the time required for the entry process depends on the exploited pathway [6]. 

Since the endocytic pathway involves many complex mechanisms, it is slower than the direct fusion between the plasma membrane and the viral envelope. Consequently, interfering with the latter resulted in a higher inhibitory effect than the one obtained by the inhibition of the endocytic pathway. However, only hampering both entry mechanisms led to complete protection from viral infection in Calu-3. In contrast to the other two cell lines, Calu-3 allowed the evaluation of both entry pathways. These data suggested that infection of Calu-3 looks similar to the entry of SARS-CoV-2 during natural infection, where the virus could differently exploit both entry mechanisms depending on its biological features and the site of the viral replication.

Point mutations, insertions and deletions were documented in the entire viral genome, especially in the S protein for its exposure to selective pressure. S-gene sequencing is the key for SARS-CoV-2 surveillance, as mutations could impact the different transmissibility of the viral variants or their escape capability from neutralizing antibody response [26,27,28].

From the analysis of replication kinetics of SARS-CoV-2 variants in the selected cell models, we observed that the Alpha variant infects the three cell lines with the same efficiency. On the contrary, our results confirmed the rapid replication of the Delta variant in Calu-3 and Vero E6/TMPRSS2 [29] while being extremely slow in Vero E6 cells. D614G, Beta, Gamma and Omicron were also slower in Vero E6 cells. However, although for the first variant, the replication kinetics between engineered cells and Calu-3 was similar, for Beta and Omicron, we observed that they were much faster in Vero E6/TMPRSS2 than in Calu-3. Little is known about the possibility that the VOCs differently exploit the two viral entry mechanisms. 

To deeply characterize the observed differences in the replication kinetics of viral variants, we investigated the inhibitory effect of either BFLA-1 or camostat mesylate in Calu-3 cells. Since our data suggested that SARS-CoV-2 uses only one entry mechanism in both Vero E6 and Vero E6/TMPRSS2 cells, the evaluation of the VOC entry mechanism was performed only on the in vitro model that most resembled the physiological target of the infection. We analyzed how the two inhibitors interfered with the entry process of the Alpha, Beta, Gamma, Delta and Omicron variants. 

In contrast to what was previously reported [10,30], we observed that both compounds equally interfered with the entry process of the Omicron variant, whereas infection from all the other clinical isolates was differently hampered by camostat mesylate or BFLA-1. This characterization underlined the importance of having a model in which the virus could enter host cells through both entry mechanisms. Such model will allow the evaluation of the entry mechanisms exploited by different SARS-CoV-2 variants, which are likely to be those used by the virus during the in vivo infection.

The surveillance and the isolation of viral variants are not only crucial to analyze their global spread and biological features, they also allow determination of whether plasma from vaccinated individuals can neutralize circulating SARS-CoV-2 VOCs. We analyzed the neutralizing capability of sera from eight subjects, selected from a panel of well-characterized samples [13] against viral variants of epidemiological interest. This evaluation was performed not only on the gold standard model (Vero E6) but also on the other two characterized cell lines. The reference methods to assess the neutralizing activity of sera against SARS-CoV-2 infection are those relying on the CPE evaluation, which include the microneutralization assay [13,27,31,32,33,34,35]. However, virus infection cannot be assessed through CPE evaluation in the Calu-3 model; therefore, the detection method had to be different.

We tested the Alpha variant because, since its identification in the middle of 2020, it raised concern for its increased transmissibility and rapid spread. Secondly, we assessed the Beta variant: it presents mutations that allowed an easier interaction with ACE2, and it seems to be less sensitive to neutralizing antibodies [27,36]. Last, we included the Omicron variant, which emerged at the end of 2021 and quickly spread across the world causing the displacement of the Delta variant. It has been demonstrated that Omicron’s high number of S mutations lead to partial immune evasion from even polyclonal antibody responses, allowing frequent re-infection and vaccine breakthroughs [11]. Unfortunately, it was not possible to use the Delta variant in this experimental setting for its inability to infect cells lacking TMPRSS2.

The results obtained from the assay on Vero E6 reflected the literature data: vaccine-induced antibody showed decreasing protection going from Alpha to Beta and to Omicron variants [27]. On the other hand, we observed almost no protection other than against D614G infection in engineered cells, remarkably, with zero protection against the Omicron variant. Interestingly, an opposite trend was seen using Calu-3 cells: the sera had a remarkable neutralizing effect against the Beta variant compared with against the Alpha variant. Notably, also sera with a low anti-RBD titer inhibited the Beta variant entry at both tested dilutions. Yet, even in this case, no serum except #2 was found to be capable of hampering Omicron infection. Thus, we demonstrated that the choice of an appropriate cell model can have a decisive influence on the determination of the neutralizing capacity of serum.

Then, we wanted to assess if Vero E6 and Calu-3, were comparable in dissecting the antibody response against SARS-CoV-2 infection. The impossibility of testing the Delta variant in one of the two models has already suggested how fundamental it is to choose a suitable cellular model. However, to strengthen our hypothesis, we correlated the neutralization data obtained. Different outcomes resulted in considering the four analyzed virus variants. In detail, we found that only the neutralizing data against the Alpha variant observed in Vero E6 experiments are consistent with Calu-3 results, which supports the lack of differences between the cell lines observed in the replication kinetic experiments.

The latter did not happen, however, with the remaining variants analyzed in this study. The replication kinetics of the Alpha variant in Calu-3 were slower than the Beta and Omicron variants, even if the first was more inhibited by camostat mesylate than the others. Additional experiments are required to deeply dissect this aspect and better comprehend the differences between neutralizing experiments in the two cell lines. The antibodies stimulated by the vaccination might target other epitopes of the S protein rather than the RBD, interfering with the TMPRSS2 activity or creating a steric hindrance that prevents the internalization via endocytosis. These molecular aspects deserve further studies aimed at evaluating the serum neutralizing activity in the presence of entry inhibitors.

## 5. Conclusions

A reliable in vitro model is fundamental to investigate the biological characteristics of SARS-CoV-2 variants, which could give evidence of their spread and infectivity. A better comprehension of the viral entry mechanisms could translate into new therapeutic targets to hamper the first steps of viral infection. To study these mechanisms, pseudoviruses have been widely used due to their safety and versatility. However, in our study, we demonstrated their limits to recapitulate the viral entry process in cells that expressed physiological levels of host factors.

Even though conventional cell lines are easy to handle and enable the simple study of the basics of viral infections, these models do not reflect the native tissues where the early infection stages occur. Monolayer cell cultures cannot recapitulate cellular composition, matrix complexity, tissue diversity and three-dimensional architecture. Our study could be a starting point to further investigate vaccine-induced antibody protection in a platform that provides increased similarity to the in vivo physiology while retaining the benefits of immortalized cell cultures. Even if in vitro models could only partially recapitulate the clinical and immunological features of COVID-19, they could contribute to advances in the study of this infection to identify novel tailored therapies.

As a proof of concept, our data on the antibody neutralizing activity tested on two different cell models showed significant discrepancies. Therefore, these findings suggest that using a system that considers only one of the two viral entry pathways may not fully reflect the neutralizing activity elicited in vaccinated subjects. An appropriate pre-clinical model resembling what occurs during human infection is thus needed to confirm the in vitro observations. 

An association between the nasopharyngeal expression of ACE2 and TMPRSS2 genes and the need for oxygen therapy during COVID-19 was described, underlining the importance of considering a cellular model that not only constitutively expresses the two proteins but also has levels that are the closest to the physiological condition [37]. While the serum neutralizing activity represents, thus far, the best surrogate of protection for COVID-19, it is only one of the correlates of protection from the disease, and, to complete the picture and assess the protection stimulated by vaccination, other immune mechanisms, such as the role of long-lived memory B and T cells, need to be elucidated.

## Figures and Tables

**Figure 1 viruses-14-01232-f001:**
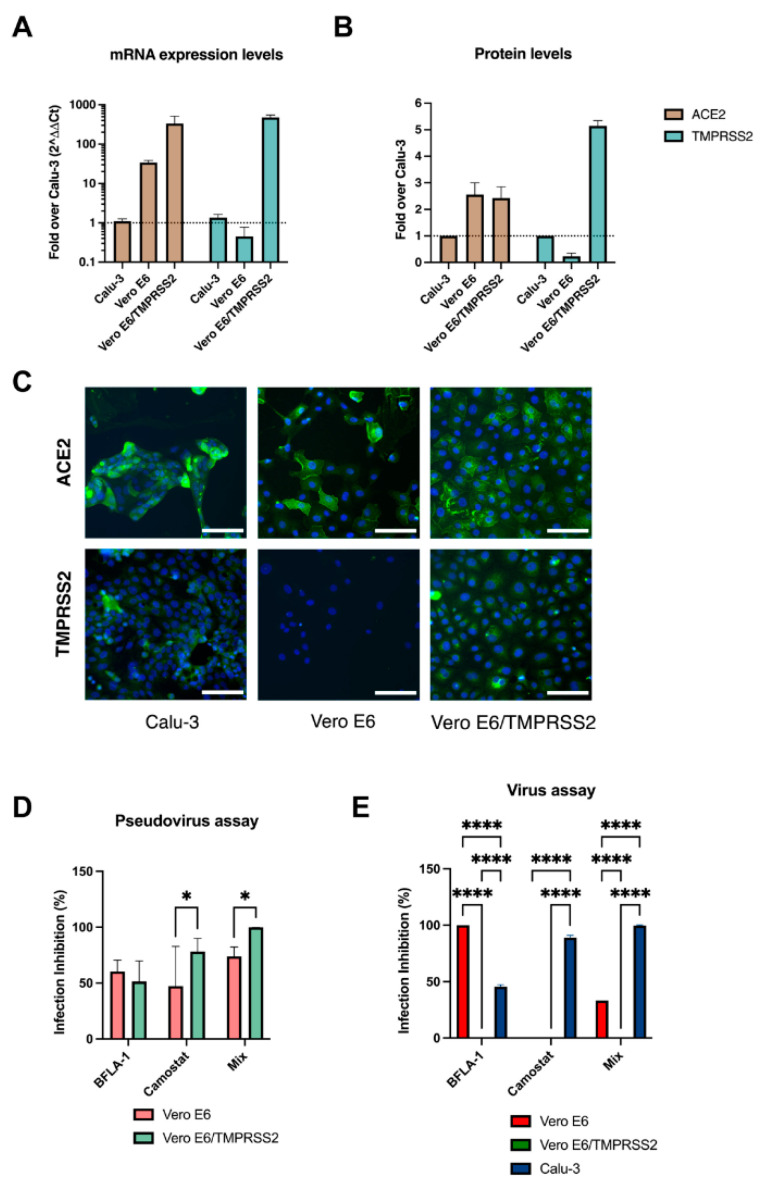
Different expression levels of ACE2 and TMPRSS2 in the different cell lines. (**A**) Gene expression was analyzed using Calu-3 as reference (dotted line). Each condition was tested in triplicate. (**B**) Immunofluorescence quantification of the protein expression in permeabilized cells. The ratio between fluorescent signal and the total number of nuclei was relatively quantified using Calu-3 as standard (dotted line). Each condition was tested in triplicate. (**C**) Calu-3, Vero E6 and Vero E6/TMPRSS2 stained for ACE2 (upper row, green signal) or TMPRSS2 (lower row, green signal). Nuclei are labelled with Hoechst 33342 (blue), scale bar, 100 μm. (**D**) BFLA-1 and camostat mesylate were used to block the entry of pseudotyped virus particles into wild type and engineered Vero E6 cells. The inhibitors were added 24 h before the transduction with pseudovirus, and the cell lysates were analyzed at 72 h. Each condition was tested six times. (**E**) D614G virus infection of Vero E6, Vero E6/TMPRSS2 and Calu-3 was selectively hampered by chemical compounds. BFLA-1 and camostat were added to the cell medium 1 h before virus infection. CPE was assessed after 48 h, and the cell supernatants were collected. Each condition was tested in triplicate. The inhibition percentages are reported as the mean values ± SD, * *p* < 0.05 and **** *p* < 0.0001.

**Figure 2 viruses-14-01232-f002:**
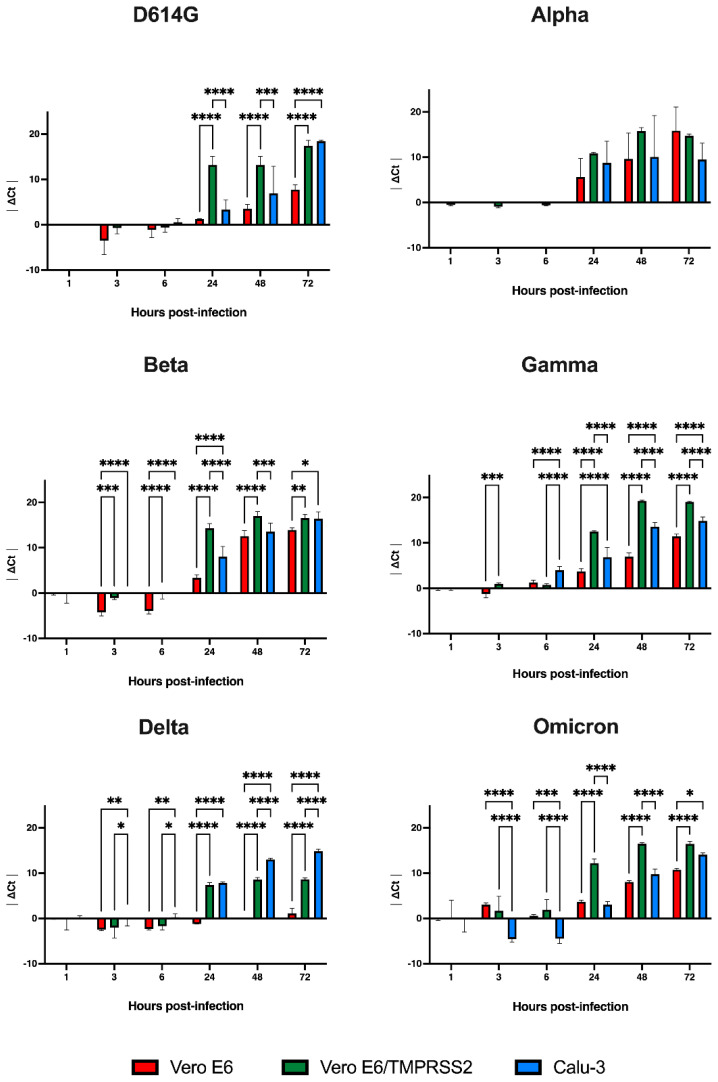
SARS-CoV-2 replication in the different cell lines. Growth curves showing the release of viral genome into the medium of cells incubated for 1 h with viruses at 0.001 MOI: D614G, Alpha, Beta, Delta and Omicron variants. The ΔCt were reported as the mean values ± SD, * *p* < 0.05, ** *p* < 0.01, *** *p* < 0.001 and **** *p* < 0.0001. Ct: threshold cycle.

**Figure 3 viruses-14-01232-f003:**
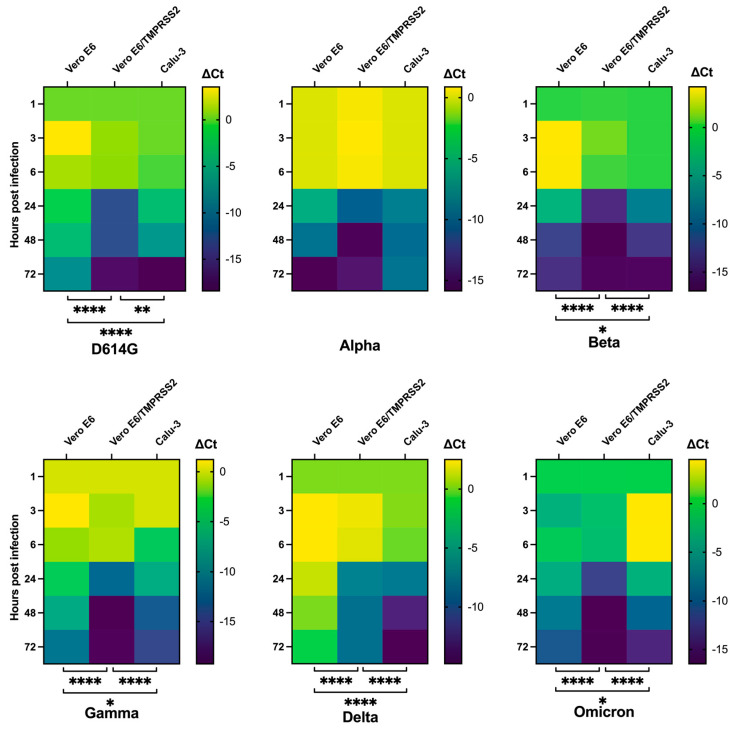
Replication kinetics trends of SARS-CoV-2 variants in the different cell lines. Heatmaps comparing ΔCt values obtained from the three cell lines at the selected time points for the tested viral variants. The ΔCt were reported as the mean values ± SD, * *p* < 0.05, ** *p* < 0.01 and **** *p* < 0.0001. Ct: threshold cycle.

**Figure 4 viruses-14-01232-f004:**
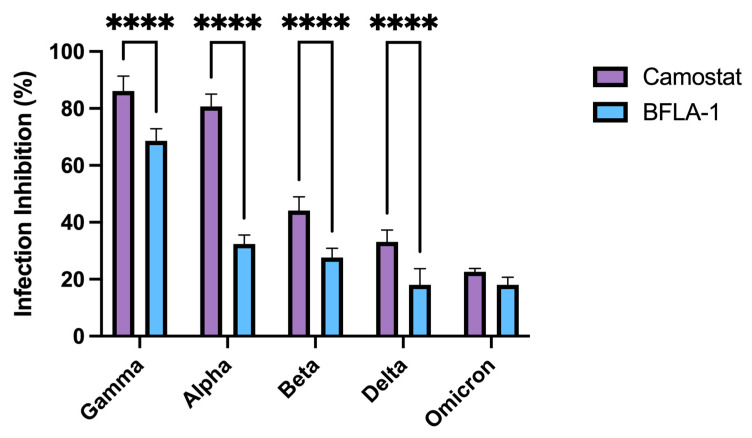
The two entry pathways differently exploited by SARS-CoV-2 viral variants to infect Calu-3 cells. Graph showing the BFLA-1 and camostat mesylate inhibitory activity against VOCs: Alpha, Beta, Gamma, Delta and Omicron. Cells were pre-treated 1 h before the infection, and the cell supernatants were collected at 72 h. Each condition was tested in triplicate, and the results were normalized over controls. The infection inhibition percentages were reported as the mean values ± SD, **** *p* < 0.0001.

**Figure 5 viruses-14-01232-f005:**
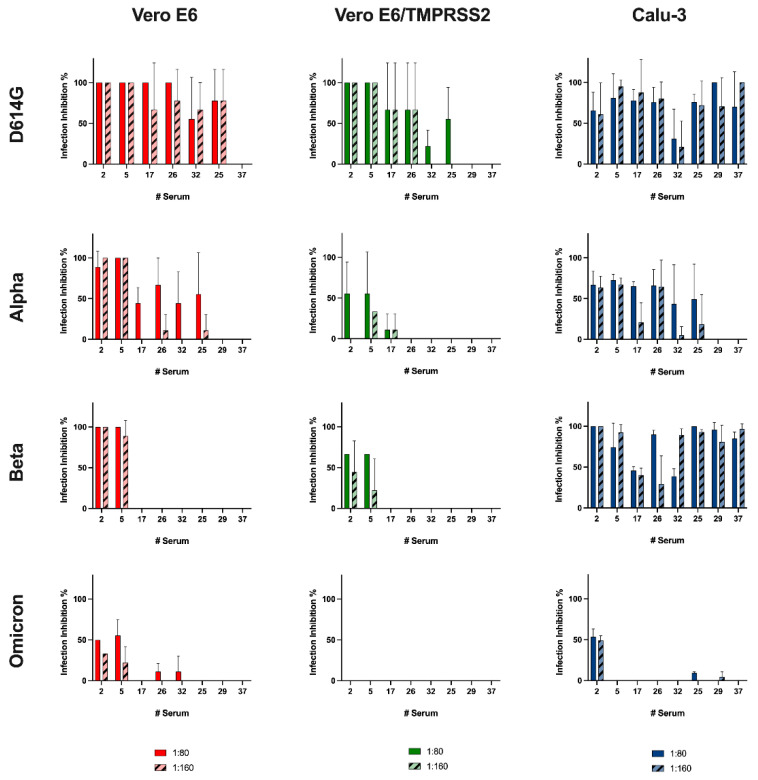
Neutralization assay using the different cell lines. Neutralization activity was assessed using two sera dilutions (1:80 and 1:160) against 0.01 MOI of four SARS-CoV-2 variants: D614G, Alpha, Beta and Omicron. Sera were incubated with the virus dilutions 1 h before the infection. CPE was assessed after 72 h, and the cell supernatants were collected. Each condition was tested in triplicate, and data were normalized over controls. The infection inhibition percentages were reported as the mean values ± SD.

**Figure 6 viruses-14-01232-f006:**
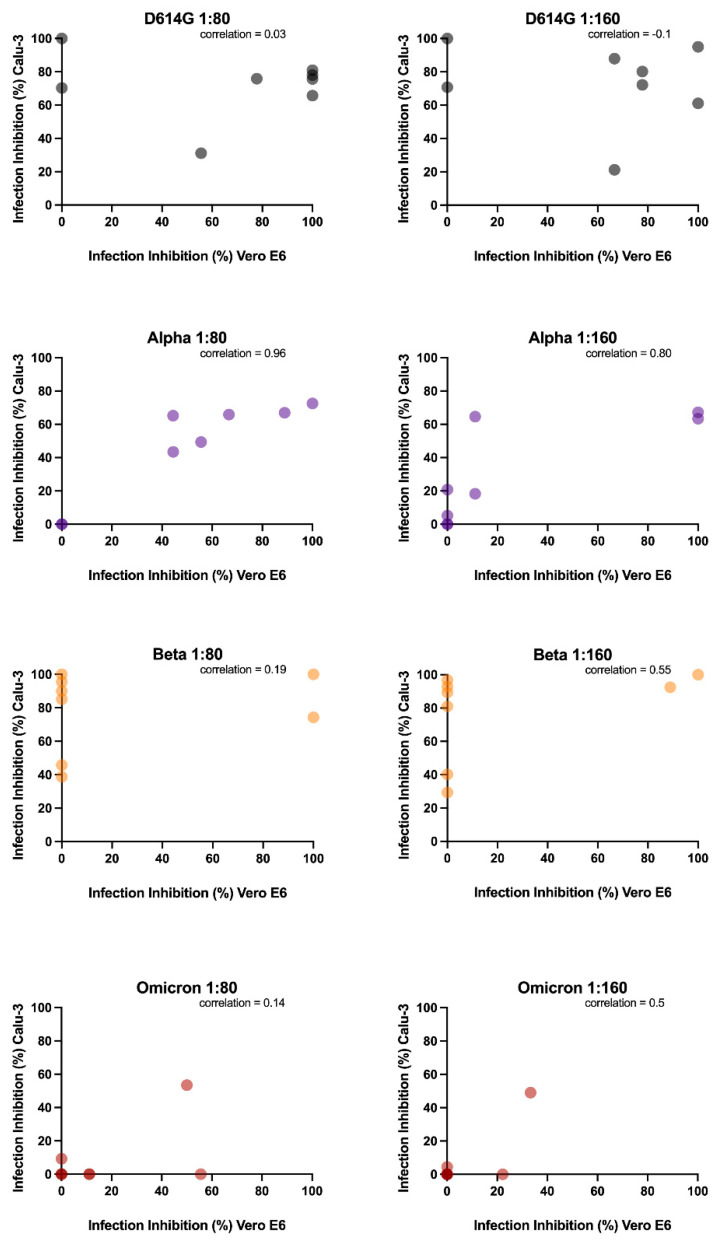
Correlation analysis. Analysis performed on the neutralization capability of the two tested dilutions of vaccine-induced antibodies on Vero E6 (*x* axes) and Calu-3 (*y* axes) against D614G, Alpha, Beta and Omicron variants.

## Data Availability

Whole genome sequence data of SARS-CoV-2 viral variants have been uploaded on GISAID database (https://www.gisaid.org/ (accessed on 3 May 2022)) with the following accession ID: EPI_ISL_413489 (D614G), EPI_ISL_1924880 (Alpha), EPI_ISL_1599180 (Beta), EPI_ISL_1925323 (Gamma), EPI_ISL_4198505 (Delta), EPI_ISL_12188061 (Omicron BA1).

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
