# Peer review of "Proper Selection of In Vitro Cell Model Affects the Characterization of the Neutralizing Antibody Response against SARS-CoV-2"

_viruses, 2022, doi:10.3390/v14061232_

Round 1

Reviewer 1 Report

In the current manuscript, Criscuolo et al. investigated the use of different cell lines to characterize the neutralizing antibody response from vaccinated sera against different SARS-CoV-2 variants. Specifically, they found that different cell lines offered different entry mechanisms for SARS-CoV-2 and thus they responded differently to neutralizing antibodies from different vaccinated sera upon infection by various SARS-CoV-2 Variants of Concern (VOCs). Overall, the manuscript is well-written, and the authors presented sufficient data to support their main claims. However, there remains some flaws in the methodology that may raise questions about their conclusions. Below are my critical comments for the present study:

Major Concerns:

  1. In Fig. 1 A, the authors measure the mRNA levels of ACE2 and TMPRSS2 genes in the three cell lines. First, there is no error bars on the bar graph. What is the n number of repeats for this experiment? Second, in the Materials and Methods section, Page 5, lines 207-225, the authors described the procedure for this experiment as “qPCR Analysis of ACE2 and TMPRSS2 expression levels”, but below (lines 222-223), they mentioned “Densitometry of GelRed-stained agarose gels”. If the authors run an agarose gel to measure the mRNA level, then it is not qPCR Analysis. Furthermore, regular PCR analyzed by Agarose gel densitometry is not a valid way to quantitatively measure mRNA level. Please perform Real-time qPCR analysis to properly assess the mRNA levels and have an n number of repeat larger than 1.
  2. In Fig. 1 B, the authors measure the protein levels of ACE2 and TMPRSS2 in the three cell lines using immunofluorescence. Immunofluorescence analysis is not the standard method in the field to measure protein level, it is more suitable to analyze protein localization and/or movement. Please perform Western Blot analysis to measure the protein levels of ACE2 and TMPRSS2 in these cell lines normalizing to some house-keeping proteins such as B-actin or GAPDH.
  3. 2, the authors presented the qRT-PCR data as just ∆Ct in which most ∆Ct values are negative. This way of showing qPCR results is very confusing to the reader and not the standard method to present qPCR data in the field. Please use the 2^(-∆∆Ct) methods and convert the data into relative fold-change difference.
  4. Please provide some demographic information on the HCP sera used in the study such as age, sex, ethnicities, underlying health problem, etc., if available. This information may be useful to some readers.
  5. In Fig. 5 A, the authors compared the neutralizing activity of 8 different vaccinated sera against four different variants of SARS-CoV-2 in the three cell lines. For Vero E6 and Vero E6/TMPRSS2 cells, the Infection Inhibition % was assessed by comparing CPE in the cells. However, for Calu-3 cells, the Infection Inhibition % was assessed by qRT-PCR of the virus released into the cell supernatants. These are two fundamentally different methods to assess the degree of viral infection and thus we may not be able to draw the same conclusion across all three cell lines with regard to the neutralizing capability of the sera. In Fig. 2, the authors measured the kinetic profiles of different SARS-CoV-2 variants in the three cell lines, and they were able to use the same method to measure the infection kinetic here (Page 5, lines 240-247). This approach should be used for Fig. 5 as well so that all three cell lines were subjected to the same method of measurement of viral infection.
  6. In Fig. 1 E, BFLA-1 was able to inhibit viral infection by 100% in Vero E6 cells while Camostat was completely ineffective in inhibiting viral infection. However, when the two agents were mixed, suddenly BFLA-1 significantly lost its potency against SARS-CoV-2. This was mentioned in the text, but no explanation was offered by the authors.

Minor Concerns:

  1. The source of BFLA-1 and Camostat Mesylate was not mentioned in the Materials and Methods.
  2. In Fig. 2, all the x-axis of the graphs is missing the “Hours post infection” label.
  3. Page 7, line 274, “ACE” missing “2”.
  4. Please indicate the timeframe of infection and sample collection in the Figure Legends of Fig. 1 D&E, Fig. 4, and Fig. 5.

Reviewer 2 Report

In this manuscript Criscuolo and colleagues evaluate neutralizing activity of the BNT162b2 14 mRNA vaccine-induced antibodies in different in vitro cellular models.

In the Fig 1, the authors characterized the mRNA and protein expression levels of ACE2 and TMPRSS2 and showed that endocytosis inhibitor BFLA-1 and membrane fusion inhibitor camostat have variable effects on infections of the three cell lines being examined- Vero E6, VeroE6/TMPRSS2, Calu-3. Figure 2 and 3examines the replication kinetics of different VOCs in the three cell lines. Figure 4 shows how the two entry pathways are exploited differentially by the VOCs. Figure 5 shows neutralization assay performed with the three cell lines and the correlation between Calu-3 and VeroE6 cell line which is currently the gold standard.

Overall, this was an interesting paper and a well-organized manuscript.  I have no serious concerns with the methodologies or any major issues with the manuscript in general.

However, a few minor concerns need to be addressed:

·       Can the authors comment on the ACE2/TMPRSS ratio of the three cell lines being examined? It seems likely there is a difference in Calu3 and VeroE6 TMPRSS cells. Seeing that there appears to be some correlation between that and disease severity (Rossi, Á.D., de Araújo, J.L.F., de Almeida, T.B. et al. Association between ACE2 and TMPRSS2 nasopharyngeal expression and COVID-19 respiratory distress. Sci Rep 11, 9658 (2021). https://doi.org/10.1038/s41598-021-88944-8) tit may be of some interest to readers

·       Line 305: “We then decided to discontinue the pseudovirus based assay” rather than “We then decided to leave/ abandon”

·       Figure 1E: It is interesting that a mix of both BFLA-1 and camostat only inhibits Vero E6 infection to 33% when BFLA-1 by itself results in complete protection. Can the authors comment what may be the reason for this difference?

·       The figure 5 panel needs to be enlarged. This data panel is one of the most important in the paper, but the current size makes the axes labels difficult to read and thus difficult to interpret.

Round 2

Reviewer 1 Report

The authors have adequately addressed all of my concerns. The manuscript can be accepted for publication in the current form with a minor change if possible. 

The graphs in Figure 5 should be made as big as possible since they are the most important data of the whole study.